# Genetic Variance Estimation over Time in Broiler Breeding Programmes for Growth and Reproductive Traits

**DOI:** 10.3390/ani13213306

**Published:** 2023-10-24

**Authors:** Bolívar Samuel Sosa-Madrid, Gerasimos Maniatis, Noelia Ibáñez-Escriche, Santiago Avendaño, Andreas Kranis

**Affiliations:** 1The Roslin Institute and Royal (Dick) School of Veterinary Studies, University of Edinburgh, Easter Bush Campus, Midlothian EH25 9RG, UK; 2Institute for Animal Science and Technology, Universitat Politècnica de València, P.O. Box 2201, 46071 Valencia, Spain; noeibes@dca.upv.es; 3Aviagen Ltd., Newbridge, Edinburgh EH28 8SZ, UK; gmaniatis@aviagen.com (G.M.); savedano@aviagen.com (S.A.)

**Keywords:** additive genetic variance, body weight, broiler, chickens, hen-housed egg production, temporal analysis, variance components, selection

## Abstract

**Simple Summary:**

Quantitative genetics theory postulates that genetic variance in closed populations under directional selection can considerably decrease. As genetic variance directly influences selection response, animal breeders are actively monitoring its changes over time to ensure the sustainability of their breeding programmes. Here, we evaluated three different approaches to computing variance components over a period of twenty-three years in a commercial broiler (meat-type chicken) population undergoing multi-trait selection. Our results showed that the trajectory of variance components fluctuated, but no overall decline trend was detected. In light of these findings, we discuss the implications for the long-term sustainability of broiler breeding programmes.

**Abstract:**

Monitoring the genetic variance of traits is a key priority to ensure the sustainability of breeding programmes in populations under directional selection, since directional selection can decrease genetic variation over time. Studies monitoring changes in genetic variation have typically used long-term data from small experimental populations selected for a handful of traits. Here, we used a large dataset from a commercial breeding line spread over a period of twenty-three years. A total of 2,059,869 records and 2,062,112 animals in the pedigree were used for the estimations of variance components for the traits: body weight (BWT; 2,059,869 records) and hen-housed egg production (HHP; 45,939 records). Data were analysed with three estimation approaches: sliding overlapping windows, under frequentist (restricted maximum likelihood (REML)) and Bayesian (Gibbs sampling) methods; expected variances using coefficients of the full relationship matrix; and a “double trait covariances” analysis by computing correlations and covariances between the same trait in two distinct consecutive windows. The genetic variance showed marginal fluctuations in its estimation over time. Whereas genetic, maternal permanent environmental, and residual variances were similar for BWT in both the REML and Gibbs methods, variance components when using the Gibbs method for HHP were smaller than the variances estimated when using REML. Large data amounts were needed to estimate variance components and detect their changes. For Gibbs (REML), the changes in genetic variance from 1999–2001 to 2020–2022 were 82.29 to 93.75 (82.84 to 93.68) for BWT and 76.68 to 95.67 (98.42 to 109.04) for HHP. Heritability presented a similar pattern as the genetic variance estimation, changing from 0.32 to 0.36 (0.32 to 0.36) for BWT and 0.16 to 0.15 (0.21 to 0.18) for HHP. On the whole, genetic parameters tended slightly to increase over time. The expected variance estimates were lower than the estimates when using overlapping windows. That indicates the low effect of the drift-selection process on the genetic variance, or likely, the presence of genetic variation sources compensating for the loss. Double trait covariance analysis confirmed the maintenance of variances over time, presenting genetic correlations >0.86 for BWT and >0.82 for HHP. Monitoring genetic variance in broiler breeding programmes is important to sustain genetic progress. Although the genetic variances of both traits fluctuated over time, in some windows, particularly between 2003 and 2020, increasing trends were observed, which warrants further research on the impact of other factors, such as novel mutations, operating on the dynamics of genetic variance.

## 1. Introduction

Breeding programmes strive for sustainable genetic progress across the traits that are part of the breeding goal. This progress depends on the parameters in the breeder’s equation, namely the selection intensity, accuracy, genetic variation, and generational interval [1]. Considerable attention has been given to the accuracy of breeding values and the prediction models used to obtain them, using either traditional (mass and pedigree-enabled) or genomic selection, when assuming that genetic parameters are stable. Conversely, studies examining the dynamics of genetic variance over long periods of time in commercial breeding programs are scarce in the research literature.

Theory postulates that genetic variation decreases under intense directional selection [1,2]. Bulmer effect build-up is expected, creating a negative linkage disequilibrium between loci and thus reducing genetic variance, and, consequently, selection response [3,4]. Genetic drift and accumulation of inbreeding in closed and finite populations also contribute to the reduction in the genetic variation. Conversely, migration, mutation, and recombination are key factors contributing to creation and shuffling of novel genetic variation [1,2,4]. The balance between the newly created variance through mutation/recombination and that lost through selection is usually unknown, while genetic variation is subject to fluctuations under constant selection [2,5]. Accounting for the dynamics of genetic variation is important since using out-of-date variances in breeding value estimation could bias the reliability of breeding values and hinder genetic gains due to the reduced additive genetic variance among selection candidates [6,7].

Computing genetic variances over time was proposed by Sorensen et al. (2001) [8]. They estimated genetic variances in cohorts consisting of distinct subsequent individuals in each cohort (distinct non-overlapping window) using a Bayesian statistical approach and a pedigree-based prediction method, the best linear unbiased prediction (BLUP). The novelty of that method was to estimate genetic parameters of a subset of data whose marginal posterior distributions were conditional on the entire data and pedigree, reducing biases from those estimations using truncated data. However, the method is computationally intensive for large datasets. Another approach is to estimate the expected variances at any time (of selection history) using the coefficient values of numerator relationship matrix **A** [9,10]. When estimating genetic variances using a pedigree-enabled relationship matrix, the estimated genetic variance refers to the variance in the base population (reference population); i.e., this estimate refers to the unrelated genetic population, not to the whole population or specific individuals in the last generation of the analysis [9]. Thus, using distinct non-overlapping windows of data and pedigree only captures part of the selection history and fragmented genetic variance changes, while consecutive, overlapping and sliding windows can provide a continuous analysis of changes in genetic variance over time. This approach requires that windows have enough data to avoid or mitigate biases of estimation [11]. Furthermore, computing the weight of factors like drift and selection causing the changes in genetic variances has been proposed, using a Bayesian approach in a dairy sheep breeding programme [12].

The estimation of genetic parameters is sensitive to sample size (number of individuals with phenotype), and large datasets are needed to ensure the accuracy of the estimation and to accurately identify changes in the genetic parameters. Datasets from chicken breeding populations have a sufficiently large size to monitor long-term genetic variances. For broiler (meat-type) chickens, body weight (BWT) and hen-housed egg production (HHP) are part of a broad and balanced breeding goal and represent the typical antagonism between growth and reproduction through a negative genetic correlation between them [6,13]. In an attempt to detect changes in the genetic variances of these two key traits, we assessed three approaches using a large broiler pedigree dataset: (a) sliding overlapping windows, (b) expected variances using coefficient values from the full pedigree in each overlapping window, and (c) a novel straightforward, practical approach that we term “double trait covariance” in the current article.

## 2. Materials and Methods

### 2.1. Dataset

The population analysed in this research comes from a maternal grandsire broiler line that has not undergone any introgression process, standing for a closed population. BWT and HHP records were provided by Aviagen Ltd. (Newbridge, UK) from a pedigreed line spanning 23 years (from 1999 to 2022). The selection was performed from overlapping generations (a total of 39 generations). BWT was recorded at 35 days of age, whilst HHP consisted of the total number of eggs collected between the 28th and 54th weeks of age. The final dataset included 2,059,869 animals with phenotype information and 2,062,112 animals in the pedigree. The data were divided into sliding overlapping windows, with a period of 3 years and an overlap of 2 years between subsequent windows. The descriptive statistics are outlined in Table 1.

### 2.2. Multivariate Model and Computation of Variances

A bivariate model was used in the analysis to estimate all variance components, as follows:$$\left[\begin{array}{c} y_{1}\\ y_{2} \end{array}\right]=\left[\begin{array}{cc} X_{1} & 0\\ 0 & X_{2} \end{array}\right]\left[\begin{array}{c} b_{1}\\ b_{2} \end{array}\right]+\left[\begin{array}{cc} Z_{1} & 0\\ 0 & Z_{2} \end{array}\right]\left[\begin{array}{c} u_{1}\\ u_{2} \end{array}\right]+\left[\begin{array}{cc} W_{1} & 0\\ 0 & W_{2} \end{array}\right]\left[\begin{array}{c} c_{1}\\ c_{2} \end{array}\right]+\left[\begin{array}{c} e_{1}\\ e_{2} \end{array}\right]$$
where the subscripts **1** and **2** represent the BW and HHP traits, respectively; y is the vector of observations; $b_{1}$ is one systematic effect, only for BW, consisting of hatch, week-contemporary group, season, sex, pen; $b_{2}$ is one systematic effect, only for HHP, consisting of hatch effect (pen–group–season); u is the vector of genetic effects (additive); c is the vector of maternal permanent environmental (MPE) effects; and e is the vector of residual effects. X, Z, and W represent the incidence matrices for systematic effects, genetic effects, and MPE effects, respectively.

Data were assumed to be conditionally distributed as
$$\left[\begin{array}{c} y_{1}\\ y_{2} \end{array}\right]\;|\;b_{1},b_{2},u_{1},u_{2},c_{1},c_{2,}R=N\left(X\left[\begin{array}{c} b_{1}\\ b_{2} \end{array}\right]+Z\left[\begin{array}{c} u_{1}\\ u_{2} \end{array}\right]+W\left[\begin{array}{c} c_{1}\\ c_{2} \end{array}\right],R\right)$$
where R is the residual (co)variance matrix between the two traits.

The (co)variances were assumed to be
$$V=Var\;\left(\begin{array}{c} u\\ c\\ e \end{array}\right)=\left(\begin{array}{ccc} A⊗G_{0} & 0 & 0\\ 0 & I_{m}⊗C_{0} & 0\\ 0 & 0 & I_{n}⊗R_{0} \end{array}\right)$$
where A is the relationship matrix of the same order as the number of animals in the pedigree; $I_{m}$ is an identity matrix of the same order as the number of levels of MPE effects; and $I_{n}$ is an identity matrix of the same order as the number of records. $G_{0}$, $C_{0}$, and $R_{0}$ are the 2 × 2 additive genetic, MPE, and residual (co)variance matrices between the two traits. All random effects were assumed to be independent.

### 2.3. Sliding Overlapping Windows

The estimation procedure, under a frequentist method, was implemented in *ASReml* [14] and convergence was assessed after maximum of 30 iterations. Parameters of a total of 22 sliding overlapping windows were estimated. To perform a Bayesian analysis, the programs *RENUMF90*, *GIBBS2F90*, and *POSTGIBBSF90* from the *BLUPF90* family programs [15] were used to obtain the marginal posterior distributions using the Gibbs sampling algorithm. One sample every 80 iterations was saved to avoid the high correlation between consecutive samples, from a chain length of 670,000 and a burn-in of 30,000 iterations.

### 2.4. Expected Variance Components

The estimations of (co)variance components of a closed population, under directional selection, pertain to the base population or the first population before the selection process starts. The same method used in the sliding overlapping windows was carried out, but under a chain length of 112,400. Expected variances were computed according to the procedure described by Legarra (2016) [9], in which all genetic variance components of a specific population can be computed at ***t*** time by partitioning the elements of an entire relationship matrix **A** into a set of individuals belonging to the population at ***t*** time. In a nutshell, variance components come from individuals that pertain to each overlapping window in the current study. Let $Var\left(a\right)={A\sigma }_{a}^{2}$ and ***t*** be the time for an overlapping window; then, the expected variances are obtained from the following equation:$${\hat{V}}_{a_{t}}=\left(\;\bar{diag\left(\;A_{t}\right)}-\bar{A_{t}}\right)*\;{\hat{\sigma }}_{a}^{2}$$
in which ${\hat{V}}_{a_{t}}$ is the expected genetic variance of an overlapping window, $\bar{diag\left(\;A_{t}\right)}$ is the average value of the diagonal of matrix **A** from only animals of the overlapping window, $\bar{A_{t}}$ is the average value of all coefficients, in matrix **A**, from only animals of the overlapping window, and ${\hat{\sigma }}_{a}^{2}$ is the genetic variance estimate using all animals from the full period (1999–2022). The differences between the variance from the base population and the expected variance are deemed the loss of genetic variance due to drift (coancestry). Differences between the expected variance and estimates from the overlapping window procedure are also expected. These differences turn into the reduction in genetic variances due to the selection process; specifically, they are because of the Bulmer effect and the preselection of animals [12]. The inbreeding coefficients were computed using the “*PedModule*” of *JWAS* Software v.1.2.1 [16] and the average of matrix **A** was computed using *Colleau’s Algorithm*, programmed in a custom Julia script (see Appendix B). *Colleau’s Algorithm* is an indirect method allowing for the computation of the average relationship ($\bar{a}$) of each member pertaining to each overlapping window, as $\bar{a}=x^{\prime }Ax$. Here, **A** stands for the full relationship matrix and ***x*** stands for a vector containing 1 in the positions of each member in the window and 0 for the rest of the individuals in the pedigree [17]. By decomposing the matrix **A**, as $\bar{a}=x^{\prime }TDT\prime x$ (where **T** stands for a lower triangular matrix and **D** represents an inbreeding-diagonal matrix [17,18]), the computation is made more straightforward than creating a dense matrix **A** by using a direct method, with the latter highly time-consuming.

### 2.5. Doubling Traits to Assess Covariances and Correlations

As an alternative approach to assess the changes in genetic variance over time, we evaluated two consecutive non-overlapping windows (of 3 years) while assuming the traits of the second window to be different and computing the covariance and genetic correlation among the four traits (i.e., a four-trait model or quadrivariate analysis). A total of 19 such analyses were carried out to examine the changes in the genetic parameters and particularly the genetic correlations between BWT_1_ and BWT_2_ and between HHP_1_ and HHP_2_. For instance, when we compared the window *“2006”* vs. *“2009*”, we were using the years 2006, 2007, and 2008 in *“2006”* (for BWT_1_ and HHP_1_), and the years 2009, 2010, and 2011 in *“2009”* (for BWT_2_ and HHP_2_). The only link between the two windows was their relationship (pedigree). In the current study, we used the correlation for the same trait to indirectly infer likely changes in its genetic variance in an empirical approach, given the number of data, and generations [19]. We deemed that a correlation greater than or equal to 0.90 indicated no relevant changes in the additive variance (steady), from 0.89 to 0.75 was a high level of steadiness, 0.74 to 0.50 was a moderate level, and less than 0.50 pointed to a strong change in the variance components between the two subsequent overlapping windows.

## 3. Results

Herein, the main results are shown graphically to reveal the trends of each variance component (Figure 1, Figure 2, Figure 3, Figure 4, Figure 5 and Figure 6).

### 3.1. Genetic (Co)variances, Heritability, and Correlation Genetics

In the current study, genetic variances and parameters were estimated using two methods: frequentist and Bayesian statistics, under the overlapping window approach. Figure 1 shows the results of genetic variance estimations, that is, the Bayesian posterior means and the highest posterior density region at 95% (HPD_95%_), and the mean and confidence intervals at 95% from using REML (frequentist). The estimates of genetic variance from REML, for the first (1999–2001), half (2009–2011), and end (2020–2022) year intervals, were 82.84, 109.85, and 93.68 decagrams squared for BWT and 98.42, 152.39, and 109.04 eggs squared for HHP, respectively. The Bayesian estimates were 82.29, 109.87, and 93.75 decagrams squared for BWT, and 76.68, 129.58, and 95.67 eggs squared for HHP, respectively. Genetic (co)variances between BWT and HHP were −48.77, −79.20, and −81.52 when using REML, and −17.86, −31.69, and −43.33 in the Bayesian analysis, respectively. Note that (co)variance estimates were almost double in REML those in the Bayesian analysis; in fact, HPD_95%_ was wider for the estimates involving HHP. Despite that, the effective size of the sample, autocorrelation lag, and Geweke diagnostic ensured the convergence of chains.

The heritability estimates of BWT were practically the same when using REML and Bayesian analysis (Figure 2a). The estimates of the first (1999–2001), half (2009–2011), and end (2020–2022) year intervals were 0.32, 0.42, and 0.36, respectively. Conversely, the heritabilities for HHP were 0.21, 0.24, and 0.18 when using REML, and 0.17, 0.20, and 0.15 in the Bayesian analysis, respectively (Figure 2b). The heritability of BWT tended to increase slightly, but it dropped again for the last overlapping window (2020–2022). However, bearing in mind the three above-mentioned periods, the changes in the heritabilities seemed to be small, especially for HHP. Moreover, over the whole 23-year period, the genetic variance for both traits tended to increase and there was no evidence of reductions in heritability irrespective of the estimation method (Figure 1 and Figure 2). Genetic correlation between BWT and HHP fluctuated over time; when using REML, its estimates were −0.24, −0.28, and −0.45, and in the Bayesian analysis, they were −0.14, −0.22, and −0.45, respectively (Figure 3). The HPD_95%_ of the latter results was wide, in accordance with most HHP estimates.

**Figure 1 animals-13-03306-f001:**
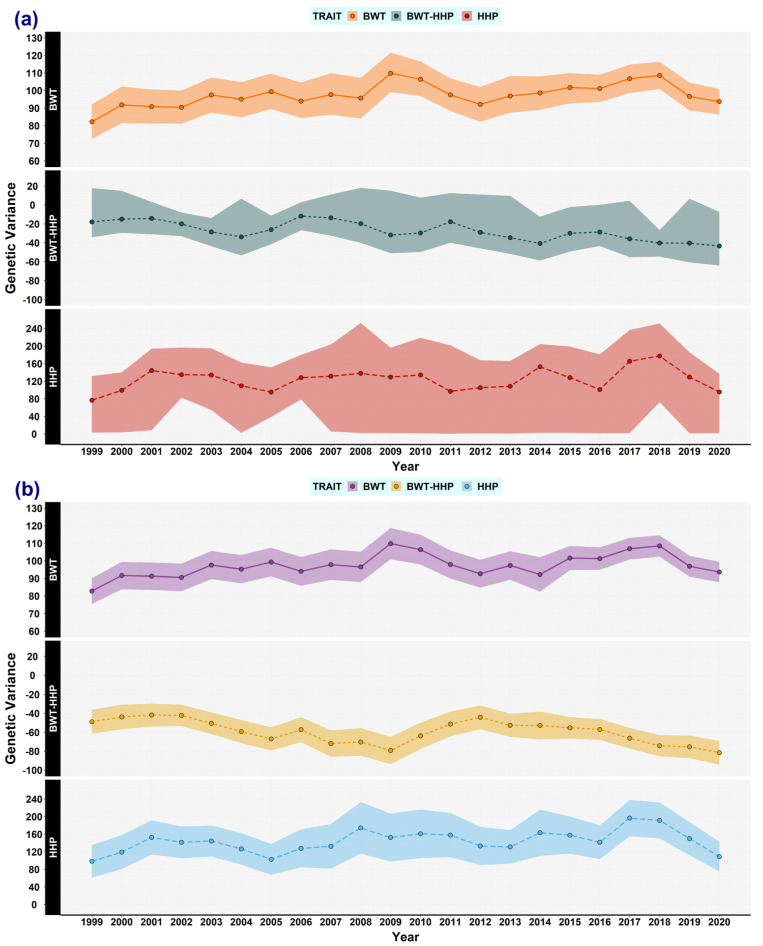
Genetic (co)variance estimates for body weight trait (BWT) and hen-housed egg production (HHP) over time. The estimates of BWT variance (solid line), BWT-HHP covariance (dashed line), and HHP variance (long-dashed line) were computed by using the following methods: (**a**) Bayesian analysis: BWT (orange), BWT-HHP (green), and HHP (red). The shaded area stands for the highest posterior density region at 95% (HPD_95%_). (**b**) Restricted maximum likelihood (REML): BWT (purple), BWT-HHP (yellow), and HHP (light blue). The shaded area stands for the confidence interval at 95%. Every dot represents an estimated mean for a window encompassing 3 years of data, overlapping 2 years with the flanking windows, from 1999 to 2022.

**Figure 2 animals-13-03306-f002:**
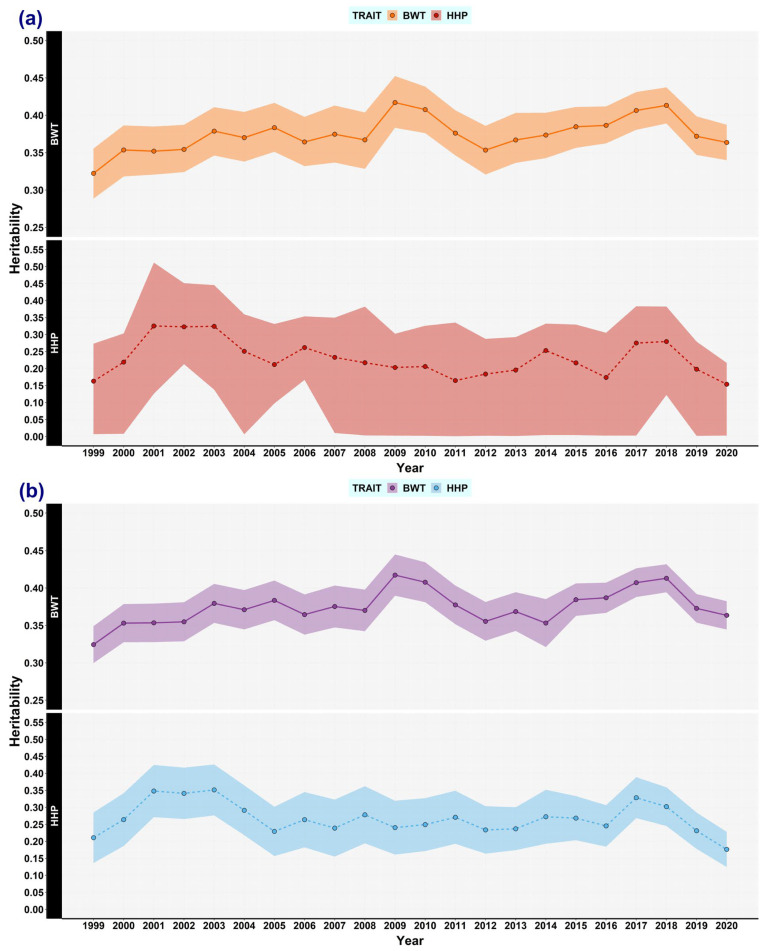
Heritability estimates for body weight trait (BWT) and hen-housed egg production (HHP) over time. The estimates of BWT heritability (solid line), and HHP heritability (dashed line) were computed by using the following methods: (**a**) Bayesian analysis: BWT (orange) and HHP (red). The shaded area stands for the highest posterior density region at 95% (HPD_95%_). (**b**) REML: BWT (purple) and HHP (light blue). The shaded area stands for the confidence interval at 95%. Every dot represents an estimated mean for a window encompassing 3 years of data, overlapping 2 years with the flanking windows, from 1999 to 2022.

**Figure 3 animals-13-03306-f003:**
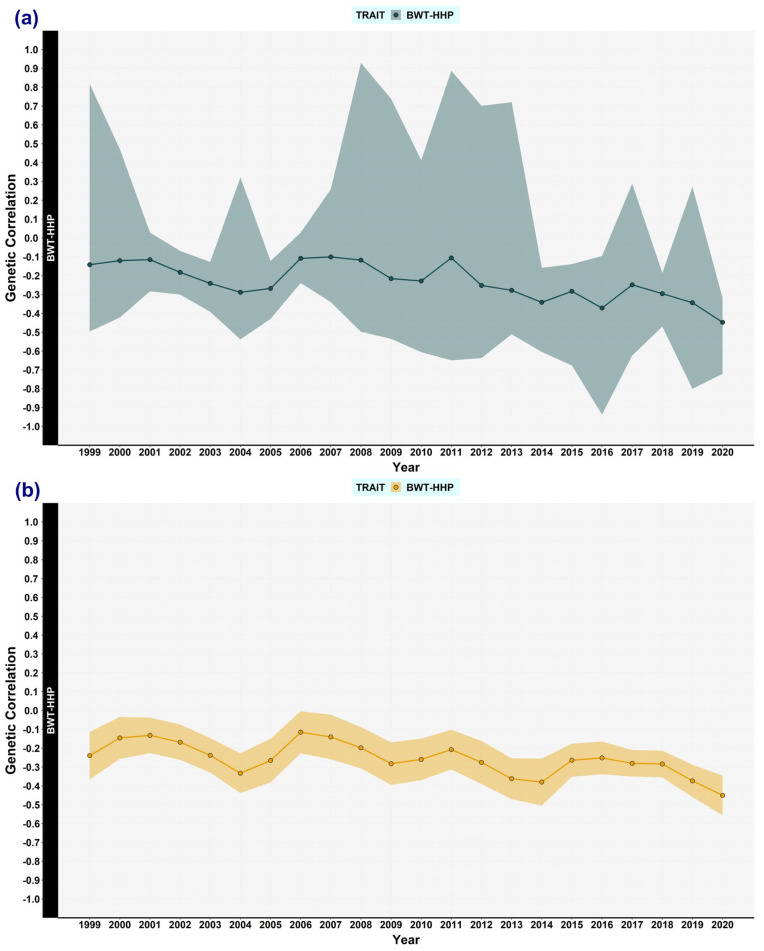
Genetic correlation estimates between body weight trait (BWT) and hen-housed egg production (HHP) over time. The estimates of genetic correlation between the two traits were computed by using the following methods: (**a**) Bayesian analysis, in green colour. The shaded area stands for the highest posterior density region at 95% (HPD_95%_). (**b**) REML in yellow colour. The shaded area stands for the confidence interval at 95%. Every dot represents an estimated mean for a window encompassing 3 years of data, overlapping 2 years with the flanking windows, from 1999 to 2022.

### 3.2. Maternal Permanent Environmental Effects

The variance component provided by MPE effects was also analysed. Its estimates for the first (1999–2001), half (2009–2011), and end (2020–2022) year intervals when using REML were 9.52, 11.27, and 6.05 for BWT, and 3.81, 23.08, and 6.53 for HHP, respectively. With the Bayesian approach, the estimates were 9.63, 11.36, and 6.53 for BWT, and 15.88, 32.74, and 12.64 for HHP, respectively. After 2017, the MPE variance dropped, registering lower values (Figure 4). This agreed with the low proportion of phenotypic variance explained by MPE in the last overlapping window (2020–2022), 0.02 for BWT, and 0.01 (REML) and 0.02 (Gibbs) for HHP. Moreover, the MPE (co)variance between the two traits tended to be zero, and thus, it could have been excluded from the model. Appendix A show the estimation of the MPE correlation, and the ratio of MPE that accounts for its contribution to the phenotypic variance, respectively.

**Figure 4 animals-13-03306-f004:**
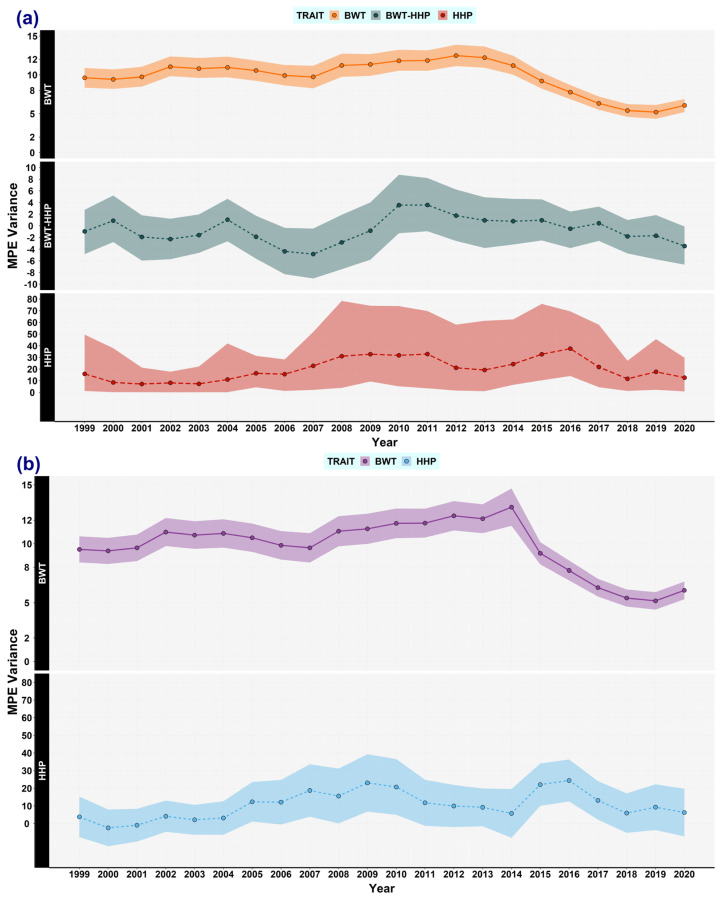
Maternal permanent environmental (MPE) (co)variance estimates for body weight trait (BWT) and hen-housed egg production (HHP) over time. The estimates of BWT variance (solid line), BWT-HHP covariance (dashed line), and HHP variance (long-dashed line) were computed by using the following methods: (**a**) Bayesian method: BWT (orange), BWT-HHP (green), and HHP (red). The shaded area stands for the highest posterior density region at 95% (HPD_95%_). (**b**) REML: BWT (purple) and HHP (light blue). The shaded area stands for the confidence interval at 95%. Most estimates of MPE covariance were around zero when using REML; hence, this component could have been fixed to zero. Every dot represents an estimated mean for a window encompassing 3 years of data, overlapping 2 years with the flanking windows, from 1999 to 2022. Note that BWT-HHP covariance when using REML is not shown.

### 3.3. Residual and Phenotypic Variances

The residual and phenotypic variances presented the same pattern in both REML and the Bayesian analysis. The Bayesian posterior means of residual variances were 163.29, 142.10, and 157.92 for BWT in the first (1999–2001), half (2009–2011), and end (2020–2022) year intervals. The residual variances for BWT were steadier, whilst they tend to increase over time for HHP with values of 375.03, 470.05, and 510.62, respectively (see Appendix A for more details of residual and phenotypic variance estimations).

### 3.4. Inbreeding Effect on the Computation of Variance Components

Inbreeding effects were analysed by comparing the genetic variance results using the pedigree within each window versus an in-depth pedigree, namely an accumulated pedigree from the specific overlapping window to the base population. The results were similar in the two approaches, as is shown in Table 2. However, the last overlapping windows showed higher values when using the in-depth pedigree than the specific window pedigree due mainly to the accumulated inbreeding. For example, the 2020–2022 window presented an average inbreeding mean of 0.03 under a 3 yr pedigree, whereas the value was 0.22 under an accumulated pedigree (23 yr). Although that turned into a difference of 11 units of genetic variance in the last window (2020–2022), the results showed that HPD_95%_s for both approaches was overlapped and presented the same pattern, an increase in the genetic variance after 23 years of selection.

### 3.5. Expected Variances and the Broiler Selection Effect on Genetic Variances

In the base population, the Bayesian variance components for BWT were 82.63 ± 2.84 (±PSD (posterior standard deviation) in decagrams squared) of genetic variance, 11.91 ± 0.39 of MPE variance, and 165.31 ± 1.25 of residual variance. For HHP, the estimated genetic variance was 100.89 ± 23.56 (in eggs squared), the MPE variance was 26.75 ± 4.49, and the residual variance was 425.24 ± 11.06. The estimated covariances between the two traits were −44.44 ± 2.81 (genetic), 1.06 ± 0.67 (MPE), and −24.40 ± 2.31 (residual). According to expected variance method, the expected reduction after 23 years of selection was at most 20% of the genetic variance of the base population (Figure 5a). The inbreeding accumulated with an annual rate below 1% (Figure 5b). Due to this increase in relationship coefficients (consanguinity) over time, it was expected that we would observe some loss of genetic variance. However, overall, there was no evident loss of genetic variance due to coancestry or selection, as the estimated genetic variances from the overlapping window approach were greater than the expected genetic variance and the variance of the base population.

**Figure 5 animals-13-03306-f005:**
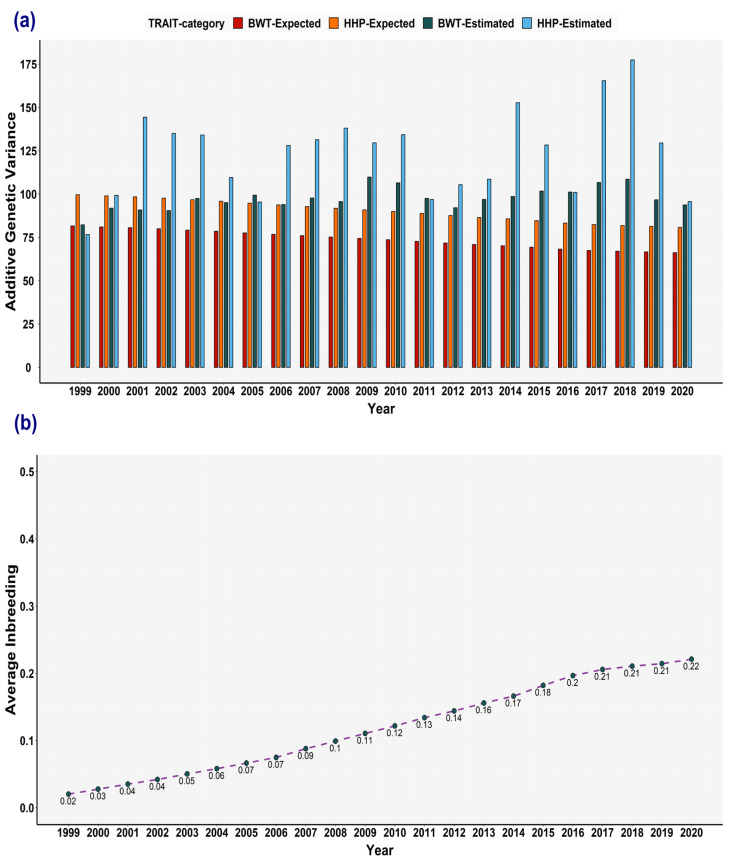
Expected variances based on average inbreeding and relationship for body weight trait (BWT) and hen-housed egg production (HHP) over time. (**a**) Expected genetic variances for BWT (red) and HHP (orange) compared to estimated genetic variances for BWT (dark green) and HHP (light blue) from overlapping window approach. (**b**) Average inbreeding for each window encompassing 3 years of data, overlapping 2 years with the flanking windows, from 1999 to 2022. The inbreeding increased progressively over time, but its increase rate was very small between windows.

### 3.6. Changes Examined by Using Double Trait Covariance Analysis

A high correlation was found between the comparisons of overlapping windows using the same trait doubled as two correlated traits. For BWT, the minimum correlation was 0.86 in the comparison of 2007 vs. 2010. Indeed, no drastic changes in BWT were detected when using double trait covariance analysis (Figure 6). For HHP, most of the estimates of genetic correlations were greater than 0.82. Nevertheless, two comparisons presented low correlation (strong change): 2003 vs. 2006 (0.33) and 2013 vs. 2016 (0.52).

**Figure 6 animals-13-03306-f006:**
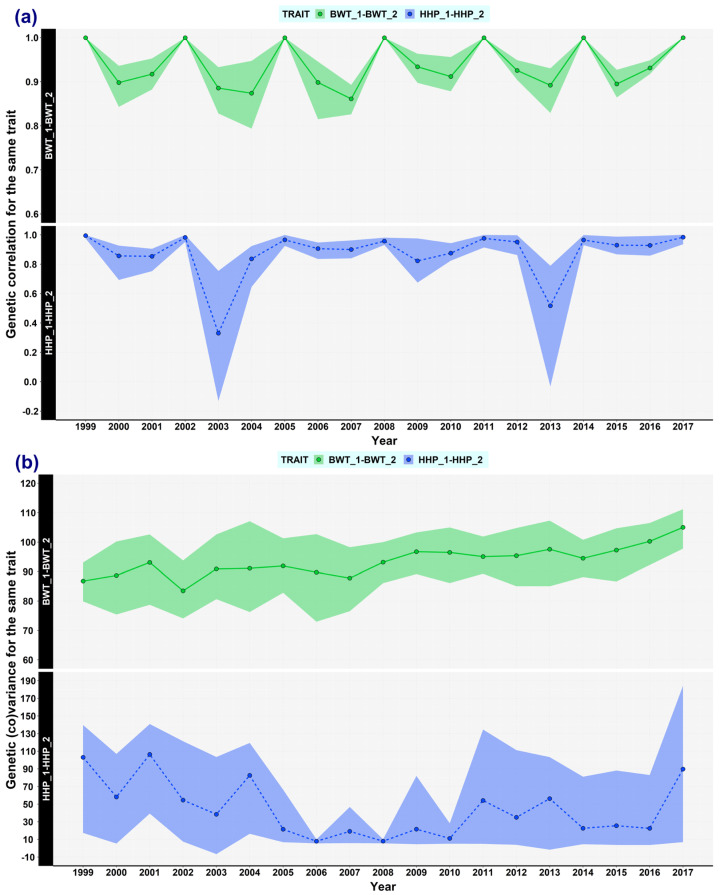
Genetic (co)variances and genetic correlations relating to the same trait for body weight trait (BWT) and hen-housed egg production (HHP) over time. The double trait covariance analysis consisted of clustering two consecutive windows without overlapping to calculate the genetic parameter for the same trait. Four traits were use in the model—BWT_1 and HHP_1 from data of the first window, and BWT_2 and HHP_2 from data of the next consecutive window—so that overlapping windows were clustered for our 19 analyses. (**a**) Results of genetic (co)variances for BWT_1: BWT_2, and HHP_1: HHP_2. (**b**) Results of genetic correlations for BWT_1: BWT_2, and HHP_1: HHP_2. The legend of the *x*-axis (Year) stands for the year of the first window. The analysis included six years (three years per window).

## 4. Discussion

### 4.1. Genetic, Inbreeding, Coancestry, and Drift Parameters

In animal breeding, accurate and unbiased estimates of variance components are essential for the prediction of breeding values. These estimates are transient and can fluctuate over time. Therefore, monitoring them and obtaining up-to-date estimations are important steps to ensure long-term genetic improvement [20].

The heritability estimate for BWT in this study (0.32) is similar to those found in commercial broiler populations at 5 weeks, of 0.320 and 0.329 in males and females [21]. These results are also consistent with previous studies in broiler lines reporting heritabilities ranging from 0.19 to 0.40 for BWT at 35 days [22,23,24,25] Heritabilities for BWT differ according to the age at recording. For instance, Chu et al. (2020) [21] reported values from 0.28 to 0.33 between the first and the sixth weeks, while other studies showed values between 0.29 and 0.40 at weeks three to six [26,27], 0.19 at six weeks in Dahlem Red chicken [28], and 0.29 at six weeks in juvenile body weight from eight overlapping generations of a broiler-type female line [29]. In some studies, sexual dimorphism for BWT was found to be significant. In commercial broiler lines, heritabilities at three different ages ranged between 0.33 and 0.40, with males (0.29–0.37) having a different range to females (0.38–0.40) [27]. When studying inter-crossed populations, BWT heritability was reported as 0.34 in F2 chickens [30] and 0.46 in an inter-cross population from two chicken lines selected divergently for BWT at 8 weeks [31].

To our knowledge, no estimations of variance components over time for antagonistic traits have been reported in commercial broiler breeding programmes, though, to date, only one study has focused on the dynamics of genetic variance of BWT over time [32]. In that research, data were used from a commercial broiler breeding population for 54 cycles of selection, but the trait definition changed over time. For the first 39 generations, BWT was recorded to a specific time of (t) days, which changed to a time of (t)-4 days for the next 7 generations, and then to a time of (t)-7 days for the last 8 generations. A Bayesian bivariate random regression model with segmented linear splines and heterogeneous residual variance was used. In the first 39 cycles of selection, the genetic variance of BWT increased, similar to our findings. In the subsequent periods, the genetic variance declined, likely due to the change in the trait definition. Unlike those results, we did not observe a reduction in BWT heritability as a result of the scaling effect [32].

The heritability estimates for HHP tended to be lower than those for growth traits. Our estimate, using all data available, was similar to those of other studies, where egg production was recorded at 40 weeks (0.11) [28], until 48 weeks of age (0.13) [29], between 1 and 17 weeks of laying (0.15) [33], and from the 3rd to the 8th month (0.125–0.184) [34,35]. In our findings, the HHP estimates were lower for the Bayesian methods when using a bivariate model than the estimates when using REML. This is primarily because Gibbs sampling is more prone to sample size effects impacting the ability to reach convergence of chains, which is particularly difficult when records are distributed unevenly [36]. In the current study, the sample size affected the accuracy of HHP estimates that showed wide HPD_95%_, given the small number of data samples and the large number of missing values compared to the records of the BWT dataset. The HHP heritabilities over time, ranging from 0.176 to 0.351 for REML and 0.154 to 0.325 for the Bayesian approach, were in agreement with those in the literature found for cumulative egg production from 3 to 6 months of laying (0.28) [37] and at 40 weeks of age in a synthetic broiler female line (0.31) [38]. Likewise, when using broiler lines with the same trait definition as in the current study, the heritability estimates were 0.24 under an additive and dominance parameters kernel [39] and 0.31 under a trivariate model [24]. None of those studies showed estimations over time for HHP. Overall, the HHP heritability estimate in this study showed a more widely fluctuating trajectory when compared with the BWT estimate (Figure 2).

Regarding the genetic covariance between BWT and HHP, its trajectory over time fluctuated; it seemed to be more stable until 2015, when it dropped, becoming more negative (Figure 1). The same trend was found for the genetic correlation (Figure 3). The last two overlapping windows showed a more unfavourable genetic correlation, which partly accounted for the reduction in genetic variance in those windows. One might expect slightly more negative correlations over time because of possible fixations of pleiotropic alleles with favourable effects on both BWT and HHP, meaning the remaining segregating alleles have more negative pleiotropism between the traits. The estimates of the first overlapping windows were near those found in other studies, between −0.18 and −0.192 [24,29], whereas the correlation was more antagonistic (−0.55) between the total number of eggs (until 17 weeks) and BWT in Thai native chickens [33], as in the last overlapping windows of the current study.

The trajectories of genetic variance for both traits fluctuated, as expected, but no overall decreasing trend was detected. The heritabilities presented a similar pattern to the genetic variance trajectories, especially for BWT (Figure 3).

The results of variances estimated with pedigree spreading over the whole period revealed a slight increase in genetic variance, which became relevant in the last five overlapping windows. That was a peculiar result since the quantitative genetic view is typically that the amount of genetic variance will be reduced by inbreeding [1,2]. However, in practice, it is still debatable whether cumulative depression effects are everlasting [10,40,41]. Furthermore, a study using maternal rabbit lines showed that inbreeding has an apparent positive effect on litter size [42], and the contribution of the dominance effect to variance increased with the inclusion of inbreeding in that model [10].

When using the overlapping windows, the changes in the variance components over time were shown. However, this approach does not identify the main causes of the changes. Estimating expected variance is a way to find out how inbreeding influences the fluctuations in genetic variance. In our dataset, annual inbreeding accumulation was lower than 1% (or 0.006 per generation), i.e., below the threshold set by the FAO [43]. Theoretically, some factors, such as drift and selection, contribute to the reduction in genetic variance in breeding programmes. For instance, the genetic variance reduction resulting from coancestry (drift) in a dairy sheep breeding program was only 3% over a period of 39 years owing to its low average relationship coefficient (about 0.002 increase per year) [12]. Therefore, a small reduction in the genetic variance coming from drift can be expected based on that study. Furthermore, the magnitude of genetic variance reduction is expected to be more significant with higher selection intensity and greater heritability [1]. For instance, in one study, growth traits with a heritability of about 0.30 experienced greater losses of genetic variances compared to fitness traits with a heritability of ≤0.11 in a commercial pig population [11]. Another factor influencing genetic variance is the presence of non-random linkage disequilibrium, also known as the Bulmer effect. It is expected to reduce genetic variance and the selection response, regardless of the magnitude of heritability, and it depends on the intensity of selection [40]. However, whilst the number of loci and population size influence long-term selection, changes in genetic parameters due to the Bulmer effect occur in early generations [44,45]. The loss of genetic variance due to drift can be computed by subtracting the genetic variance in the base population from the expected variance, while the difference between the expected variance and the estimated variance (via the sliding overlapping window method) is attributable to the loss of genetic variance due to selection. This latter effect includes both the Bulmer effect and the preselection of animals at birth based on the parent average (candidates). Figure 5 illustrates that a maximum reduction of 20% is expected due to coancestry coefficients. Nevertheless, apart from the HHP variance of the first window, all expected variances are lower than the estimated genetic variance and the variance of the base population. Reports of genetic variance reduction have been observed in commercial pigs [11], Czech Fleckvieh dual-purpose cattle [46] for growth traits, and in dairy sheep for milk production [12]. In the case of the sheep program, estimates indicated a 10% loss of genetic variance due to the Bulmer effect and preselection pressure at birth (selection) [12]. Contrary to these findings, research on variance component estimations using random regression revealed a significant increase in genetic variance for milk production in dairy cattle [19,47]. Similarly, a study in a commercial broiler population exhibited an upward heritability for BWT [32].

In practice, breeding companies control the inbreeding effect by implementing optimal contribution selection or by straightforward ad hoc restrictions on the inbreeding rate in truncation selection, e.g., no cousin mating [41] or avoiding full- and half-sibling mating. These measures can mitigate the impact of inbreeding on the genetic variance and, indirectly, the selection effect over time by maintaining a suitable effective population size (Ne) [20]. That restriction of mating between animals with high relationship coefficients and the large effective population size could account for the lack of significant reductions in genetic variance in the current study.

### 4.2. Monitoring the Dynamics of Genetic Variance

Theory postulates that the genetic (co)variance, heritability, genetic correlation, and selection response of a trait under direct selection are reduced, especially when one or a limited number of traits are considered. However, genetic variances could be less influenced when the selection is performed for multiple traits, and the selected population undergoes different selection pressures depending on each trait [12,45]. Moreover, the trait definition is updated as a response to requirements [23,32,48]. For example, BWT was recorded at 6 weeks from 1987 to 1998, and from then on, it started to be recorded at 5 weeks [23]. This kind of change could introduce a disruption in favour of or against the genetic variance of the new trait [32].

In practice, nucleus populations are continuously under selection for fitness, while family sizes are regulated to reduce variation and maintain Ne. Commercial poultry breeding goals have broadened since the 1970s, typically including 40 to 50 traits now [41]. Even, the well-known Virginia body weight lines, an experimental population of White Plymouth Rock chickens selected divergently for only BWT at 8 weeks, showed a progressive response to selection, and later on, suitable genetic variance after 60 years of selection. In that case, a strong standing genetic variance brings about the maintenance of genetic variance over time [49,50]. The standing variance is also highly linked to mutational variance [51]. Due to computational complexities, obtaining estimates for the contribution of mutational variance in large chicken datasets is challenging; however, evidence from other species suggests that de novo mutations can have an important role in the maintenance of genetic variation over time [5,52].

In multi-trait breeding programmes, several traits may have an antagonistic relationship that can affect the long-term selection response if not appropriately managed. For instance, a negative high correlation (−0.92) between the number born alive and body weight at weaning in rabbits does not enable breeders to establish a feasible line selected for both traits (instead of creating independent lines) [53]. However, in our study, the magnitude of genetic, MPE, residual, and phenotypic correlations between BWT and HHP was moderate, thus allowing for selection in the desired direction for both traits simultaneously. The persistency of antagonistic correlations relies on the genetic architecture between traits, expressed as pleiotropy with tight or loose linkages amongst QTLs [54].

Although adverse negative genetic correlation hinders responses to selection, poultry breeding schemes using broad and balanced breeding goals (multi-trait selection) have shown that favourable selection responses in antagonistic traits can be achieved [41]. A plausible explanation could be the simultaneous fixation of favourable haplotypes or QTLs for the traits involved in the antagonism [54]. This explanation could match the high apparent correlation in the last overlapping windows, becoming more negative, in the current study. In fact, we found that the genetic variance slightly dropped, likely due to a stronger selection in the last overlapping windows (from 2018), showing a higher antagonistic genetic correlation as a result of a better phenotypic response in BWT (Table 1). In pigs, remarkable reductions in genetic variance and heritability were found due to antagonistic genetic relationships between fitness and growth traits [11]. Thus, monitoring the correlation between traits is important for poultry breeding programmes. Selection for leg health traits is an important example in broiler breeding programmes. Despite moderately unfavourable genetic correlations between leg health traits and BWT, these traits can be improved simultaneously when balanced breeding goals are implemented [23].

The three proposed approaches (overlapping window estimations, expected variances, and double trait covariance analysis) are feasible strategies to monitor the dynamics of genetic variance. Estimates from overlapping windows use both phenotypic and pedigree data, rendering the approach more accurate since the expected variance method and double trait covariance analysis are based mainly on the pedigree information. The latter method enables us to corroborate the magnitude of changes in the genetic variance over time. In dairy cattle, the concern about genetic variance estimations brought about a method to approach trends in genetic variances over time and their tolerance values in the breeding programme, based on the estimates within strata using breeding values and prediction error variance (PEV) of Mendelian sampling deviations [55,56]. This method helped to recognize estimates outside the tolerance interval for milk yield, and fat yield between 1998 and 2006, and consequently they supported suitable decision-making in further genetic evaluations [57]. Furthermore, a study in Australian Holstein cattle revealed that the genetic correlations of the same trait, such as protein yield, fat yield, or some type traits, measured in different years were less than 1.0 [19], which was likely to be due to selection or changes in trait definitions [19,36]. The correlations between consecutive periods were high overall in that study, which agrees with our findings. However, some traits showed a low correlation in consecutive periods, e.g., the survival trait, with 0.21 ± 0.63 (mean ± SE). In our study, the comparison between the consecutive windows 2013 and 2016 showed the lowest correlation (0.33) for HHP. This change, apparently positive, in HHP genetic variance could have multiple explanations but those are beyond the scope of this study. The double trait covariance method seems to be practical only in non-overlapping consecutive windows since comparing distinct windows (non-consecutive) can bring about convergence issues [19].

On the other hand, large datasets and bias are key factors to consider when estimating variance components, and estimates seem more susceptible when the selection includes further genomic information [58,59]. In previous research, the heritabilities in pigs were smaller and genetic correlations were greater by using genomic data than when using estimates with only pedigrees and phenotypes [11]. In broilers, the genetic correlation estimates changed gradually from a negative value when using only the pedigree (−0.192) to a value close to zero with a full genomic kinship matrix for BWT and HHP [24]. Ignoring the previous selection process for each window could also increase the bias. However, recent studies demonstrated that tracing back three generations in the pedigree and removing animals not contributing own or progeny phenotypes increases computational efficiency without changing the ability to predict breeding values, thus reducing bias [60]. This depends on the number of years or generations used to estimate variance components. In practice, in poultry breeding companies, the variance components are usually estimated in a cohort of three years, as in the current study. The windows included an average of 10 generations to attain computational efficiency, as the size of the data window creates a trade-off between modelling fast changes and bias due to the use of truncated data [11,40]. Thus, we expected that the level of bias would not dramatically affect the result of the sliding overlapping window method.

Monitoring the genetic variance is a key factor to be considered in poultry breeding programmes to assess breeding sustainability. The three approaches shown in the current study can highlight changes or loss in genetic variances. Overall, the results, in agreement with other studies, suggested that little variation was lost, implying that rates of genetic improvement can be sustained in the future [20].

## 5. Conclusions

Monitoring genetic variance in the broiler breeding industry is an excellent strategy to ensure positive levels of genetic progress. The methods presented in this study can be extended to other traits in the breeding goal, provided datasets are large enough. Although genetic variances fluctuate over time, they may sometimes strikingly increase. In the current study, we found this happened particularly between 2003 and 2020, indicating suitable levels of maintenance of genetic variance. A suggestion is that other factors, such as a sufficiently high rate of de novo variation (i.e., mutations), can counterbalance the loss of variation, which warrants further studies to elucidate these factors contributing to novel genetic variance.

## Figures and Tables

**Table 1 animals-13-03306-t001:** Descriptive statistics of the dataset included in the analysis per window (BWT: body weight at 35 days; HHP: hen-housed egg production).

Overlapping Window	Pedigree ^1^	Number of Generations ^5^	BWT ^2^			HHP ^4^		
			Number ^3^	Mean	SD	Number ^3^	Mean	SD
1999–2001	209,977	9	207,734	178.11	22.62	4711	108	22.46
2000–2002	221,795	9	219,618	179.10	22.34	4544	111	21.43
2001–2003	231,672	10	224,297	178.80	22.26	4908	113	21.17
2002–2004	229,175	11	226,534	179.90	23.06	5126	115	21.42
2003–2005	232,122	10	230,104	185.58	24.62	5166	117	21.04
2004–2006	233,058	11	230,458	192.90	24.93	4841	116	21.48
2005–2007	218,953	10	211,600	199.45	24.17	4983	115	21.01
2006–2008	203,089	11	200,852	200.77	21.13	4694	118	22.66
2007–2009	203,461	11	201,163	202.50	24.06	4732	121	24.05
2008–2010	213,624	11	211,513	203.85	24.13	4964	125	25.17
2009–2011	207,383	10	205,448	205.21	23.93	5199	125	25.30
2010–2012	214,657	11	212,462	207.21	23.77	5445	125	27.42
2011–2013	224,160	11	221,943	205.99	23.88	5652	116	31.09
2012–2014	234,220	10	231,895	205.87	24.17	6027	118	31.28
2013–2015	243,054	9	240,617	205.03	24.41	6412	119	29.55
2014–2016	288,859	10	286,084	206.12	24.37	7490	122	25.54
2015–2017	329,530	10	326,546	205.71	24.34	7465	122	24.99
2016–2018	362,340	10	359,044	205.26	23.98	7624	121	24.79
2017–2019	380,101	10	376,786	204.79	23.51	7971	123	25.03
2018–2020	361,019	9	357,614	205.41	22.86	8115	125	27.26
2019–2021	340,604	9	336,671	205.48	23.74	7159	127	28.18
2020–2022	355,976	10	351,908	210.48	25.17	5876	126	28.65

^1^ Number of animals in the pedigree file for the overlapping window. ^2^ Body weight trait in decagrams. ^3^ Number of animals with records in the phenotype file for the overlapping window. ^4^ Hen-housed egg production in egg unity. ^5^ Number of overlapping generations that are included in each overlapping window.

**Table 2 animals-13-03306-t002:** Estimates of genetic variances under pedigree within sliding overlapping window and accumulated pedigree for body weight trait.

Overlapping Window	Pedigree within Window ^1^			Accumulated Pedigree ^4^		
	Mean	HPD_95%_ Interval ^2^	IChS ^3^	Mean	HPD_95%_ Interval ^2^	IChS ^3^
2000–2002	91.91	[81.42, 102.30]	182	90.43	[80.81, 99.30]	92
2001–2003	90.89	[81.33, 100.70]	142	90.53	[81.98, 100.30]	20
2002–2004	90.51	[81.07, 100.00]	64	91.89	[82.66, 101.10]	32
2003–2005	97.52	[87.55, 107.40]	228	97.45	[88.28, 106.70]	156
2004–2006	95.12	[84.76, 104.90]	114	96.50	[84.36, 107.20]	286
2005–2007	99.39	[89.51, 109.70]	114	95.81	[86.53, 106.10]	68
2006–2008	94.02	[84.37, 104.60]	50	92.27	[79.32, 106.10]	214
2007–2009	97.73	[86.29, 109.90]	224	92.56	[83.16, 103.30]	60
2008–2010	95.70	[84.03, 107.30]	544	94.26	[83.93, 104.10]	120
2009–2011	109.87	[99.20, 121.60]	332	103.76	[90.31, 115.60]	60
2010–2012	106.49	[96.84, 116.60]	110	110.25	[99.45, 121.00]	46
2011–2013	97.58	[88.42, 107.00]	174	103.63	[89.92, 113.40]	256
2012–2014	92.15	[82.18, 102.10]	134	97.67	[86.74, 108.40]	256
2013–2015	96.93	[87.42, 108.30]	272	101.40	[89.96, 114.20]	258
2014–2016	98.68	[88.99, 108.10]	36	98.44	[84.85, 114.40]	334
2015–2017	101.74	[92.68, 110.00]	32	109.19	[98.05, 123.50]	342
2016–2018	101.18	[93.42, 109.00]	40	112.57	[98.33, 122.20]	254
2017–2019	106.75	[98.62, 114.80]	482	119.45	[110.40, 129.10]	334
2018–2020	108.63	[100.90, 116.40]	68	123.41	[107.00, 137.50]	322
2019–2021	96.71	[88.71, 104.60]	482	108.18	[97.88, 119.10]	370
2020–2022	93.75	[86.42, 100.90]	206	105.07	[94.59, 113.70]	288

^1^ Pedigree comprises only animals for the 3-year window. ^2^ Interval of the highest posterior density region at 95%. ^3^ Independent chain size. ^4^ In-depth pedigree including animals from the overlapping window to the base population.

## Data Availability

The data belong to Aviagen Ltd.; thus, they are unavailable due to privacy restrictions.

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
