# Peer review of "Genetic Variance Estimation over Time in Broiler Breeding Programmes for Growth and Reproductive Traits"

_animals, 2023, doi:10.3390/ani13213306_

Round 1

Reviewer 1 Report

This manuscript analyzes genetic variance of body weight and hen housed egg production in a commercial broiler population of over 2 million chickens, using three different methods.  For the long-term success of the poultry industry, studies like this are important.  Thank you for sharing. 

What is not clear is if there was any introgression throughout the 23 years.  It may have been stated and I missed it, but if used it should be stated for the reader to understand the full picture.  It could have contributed to the steady genetic variation.  It is also important to note when comparing to other experimental lines and could explain differences in heritability from closed populations such as the VA BW line.  

A few specific comments:

Lines 39, 230, Table 1, and elsewhere:  Be careful of "BWT" becoming "BTW"

Line 81 "intensive" typo

Line 93 sentence ends in ".," errant comma.

Line 199 greater than or equal to 0.90

Line 200 less than 0.50?

Line 164 a set of individuals (plural)

Line 390 "till"?

The Discussion section appears to have many more grammatical errors than the rest of the paper, thus I will not detail them.  That section needs to be edited by a "fresh eye" for typos, syntax, and punctuation.

The Discussion section is the area of greatest concern.  I can understand the thought processes.  However, the attention to detail is not the same as in the rest of the document (see last comment/suggestion).

Reviewer 2 Report

Review of “ Genetic variance estimation over time in broiler breeding programmes for growth and reproductive traits” by Sosa-Madrid et al.

Animals 2023, 13, x. https://doi.org/10.3390/xxxxx

Summary

This work investigates the changes in genetic (co)variances for two traits in a long-term broiler breeding line, using records on ~2 million individuals. Results show that 23 years of selection in broilers has not lead to a meaningful reduction in genetic (co)variances. The topic of this manuscript is relevant, and the current work adds important information to the literature.

Overall the study is straight-forward and mostly well executed, albeit methodologically somewhat simple and therefore approximate. At some places the writing can be condensed and repetition can be removed (particularly in Discussion). The Discussion of the Bulmer effect and of the effects of inbreeding should be improved, and consequences of using a selected generation as base population for the estimation of variance components should be discussed (How confident are you that your estimates are unbiased?). Also information on the generation interval should be included, so that the reader can judge how many generations are included in windows of 3 years.

Main  comments

The Bulmer effect reduces the between family component of the genetic variance, not the Mendelian sampling component of the genetic variance. When the data that are used to select the animals is included in the analysis, then the mixed model “sees” the selection and the ReML estimate is not reduced by selection. When we have only part of the data (e.g. a window of 3 yrs), then the model does not “know” that the first generation of the window is from selected parents (which reduces the estimate). But the model still “sees” the full Mendelian sampling variance in the generations after the first generation of the window (which works to push the estimate towards the true base generation value). Therefore, when using a window with the first generation descending from selected parents, the MME-ReML estimate is some mix of the Bulmer-effect-reduced variance and the true base generation variance. The longer the window (e.g. 6 generations instead of 3) the closer the estimate to the true base generation value; the shorter the window (e.g. 1 generation) the smaller the estimated genetic variances. But the precise mix is unknown (unless you do some mathematical derivation of it). Hence, you cannot conclude that the estimates from the current study include the reduction in variance due to the Bulmer effect, and it is difficult to conclude how much bias there will be.

L441-448: Here you state that “control of inbreeding … can mitigate the effects of inbreeding on the genetic variance and indirectly the Bulmer effect”. But OC selection decreases the rate of inbreeding, not the impact of a given rate of inbreeding on the genetic variance. You find an increase in the inbreeding level of ~0.22 in the full time period. Whether this has an effect on the genetic variance does not depend on whether this level is reached with OC selection or an other type of selection. Second, OC selection does not prevent the Bulmer effect; it has very little to do with that. I strongly suggest dropping L441-448; it does not add anything and is mostly incorrect.

Scaling effect: We often see that the variance of a trait increases with its mean, so that the coefficient of variation remains constant. If we look at the two traits in this way, does the conclusion change? (i.e. is did the coefficient of variation decrease?). This could be discussed.

L199 and other places: A high correlation between the trait measured in different windows is interpreted as “no relevant change in additive genetic variance”, which is incorrect. The correlation does not provide any information on the change in additive genetic variance, so this conclusion must be corrected.

Detailed comments

L42 “lower than estimates using overlapping windows.”

Material and Methods: Can some information be included on the trait definitions? Were they constant over the 23 years? (probably not?).

L167: “then, the expected variances under a drift model are obtained from …”

Figures: for figures showing variances, please let the y-axis start at zero (remove the negative part).

L266: The current work is not about hypothesis testing (testing whether the cov is different from zero), but about estimation. There is no reason to omit estimation of the covariance when it is not significantly different from zero. Therefore, I suggest to include a panel in Figure 4b for the covariance. Furthermore, this would also provide information on the confidence interval.

L362: Did you miss the following broiler study?  Mebratie W, Shirali M, Madsen P, Sapp RL, Hawken R, Jensen J. 2017. The effect of selection and sex on genetic parameters of body weight at different ages in a commercial broiler chicken population. Livestock Science. 204:78-87. doi:https://doi.org/10.1016/j.livsci.2017.08.013.

Figure 6b: this figure has a strange pattern with the correlation being equal to 1 once every three years. This seems an artefact?

L404-411: If the genetic variance does NOT really change over generations, and a model is fitted that uses the full 23 years pedigree, then it is expected that this model will produce higher estimates than estimating variances per window, particularly for the later windows. So the sentence “That was a peculiar result…”  is not correct, because this result is expected. The reason is as follows: the drift model (i.e. the A-matrix in the analysis) says that var(a)_t = (diag_At_avg – At_avg) * Va,0. In the data, the model “sees” a certain value for the var(a)_t, and in the relationmatrix it “sees” (diag_At_avg – At_avg). So if the genetic variance does not decrease, but the relationship matrix says that is should decrease, then this is simply compensated by a higher Va_0 estimate.

L416: the FAO guideline refers to the rate of inbreeding PER GENERATION, whereas the current estimate is per year. Hence, some translation from per year to per generation is needed to make this comparison. Please include information on the generation interval.

L427: The Bulmer effect depends on selection intensity and selection accuracy, not solely on the intensity.

L429: see above comment about Bulmer effect.

L483: “the the the”

L494: At a few places, including here, the term “balanced breeding goal” is used, which is a positive judgement (in a moral sense). I suggest to be a bit more neutral with respect to terminology, and refer to a “multitrait breeding goal” instead.

none

Reviewer 3 Report

This study investigated the changes in genetic variance of two important economic traits (growth and egg production) in a commercial broiler population of 23 years (1999-2022). The comments about this study are as follows:

1.         Individuals from three years were treated as a window, so the effect of different year should be included in the model. Otherwise, the results of this study could not be evaluated.

2.         Some more information about the animals used in this study should be described detailed, such as the number of individuals used in every generation, how many male and female birds, how many pens, hatches and etc.

3.         The estimation method of the genetic covariance between BWT and HHP is not described clearly.

4.         In the figures of this study, HHEP was used, whereas HHP was used in the text. I think it must be a mistake.

5.         “On the whole, genetic parameters tend slightly to rise over time.” Is there a significant test to describe this result?

6.         Standard errors or confidence intervals should be added to Fig. 5 to reflect error ranges of parameter estimates.

7.         The unit of BWT and HHP should be given in the table.

8.         The confidence interval of the genetic correlation in Fig. 3 was every big, why?

9.         In Table 2, the first window contains individuals from 2000 to 2002 and the last two windows are 2019-2021 and 2020-2022, the windows were not consistent with the other analysis. Is there any reason for this? Also in Fig. 6, the last window is 2017, where are the other windows?

None

Reviewer 4 Report

The paper is well presented, interesting and brings great contributions to the understanding of genetic trends in broiler breeding programmes. Some doubts and suggestions are described below:

1 - The term "trend" is mentioned several times in the text. I think it would be useful if simple trend analyses, estimating single regression trend lines, were estimated and plotted within the figures.

2 - Many theoretical arguments are presented to explain/justify genetic variability (62-68) but migration is not mentioned. I think it would be useful to mention the effects of migration on genetic variance.

3 - The authors suggest in many parts of the text (simple summary, introduction and materials and methods) that the population is closed. But I think it would be better to explicitly mention this in the Dataset section so there is no doubt. The reason for this is that immigration can change the genetic variance as it can be seen in the years 2009 and 2018 where there are peaks.

4 - 68-69: The balance between the newly created variance through mutation/recombination and that lost through selection is usually unknown.

I think mutation rates are too low to justify newly created variance and compensate for selection. Specially if selection intensity are moderate/high.

5 - Results: What can explain the peaks of genetic variance in 2009 and 2018?

226-227: However, the changes in the heritabilities seem to be negligible, especially for HHP. 

I don’t think a change of 0.1 in heritability, which means 0.1/0.32=31,25%, is negligible.

Round 2

Reviewer 3 Report

 Accept in present form